# High Mobility Group Box 1 Is Potential Target Therapy for Inhibiting Metastasis and Enhancing Drug Sensitivity of Hepatocellular Carcinoma

**DOI:** 10.3390/ijms26083491

**Published:** 2025-04-08

**Authors:** Arunya Jiraviriyakul, Chatchai Nensat, Samitanan Promchai, Yanisa Chaiaun, Yanisa Hoiraya, Nutnicha Yamnak, Suphakit Khutanthong, Nun Singpan, Worawat Songjang

**Affiliations:** 1Integrative Biomedical Research Unit (IBRU), Faculty of Allied Health Sciences, Naresuan University, Phitsanulok 65000, Thailand; arunyaj@nu.ac.th (A.J.); chatchaine@nu.ac.th (C.N.); 2Department of Medical Technology, Faculty of Allied Health Sciences, Naresuan University, Phitsanulok 65000, Thailand; 3Department of Cardio-Thoracic Technology, Faculty of Allied Health Sciences, Naresuan University, Phitsanulok 65000, Thailand; 4Biomedical Sciences Program, Faculty of Allied Health Sciences, Naresuan University, Phitsanulok 65000, Thailand; 5Medical Technology Program, Faculty of Allied Health Sciences, Naresuan University, Phitsanulok 65000, Thailand; yanisachaiaun@gmail.com (Y.C.);; 6Department of Pathology, Faculty of Medicine, Naresuan University, Phitsanulok 65000, Thailand; suphakitk@nu.ac.th (S.K.);

**Keywords:** high mobility group box 1 (HMGB1), hepatocellular carcinoma (HCC), metastasis, sorafenib, oxaliplatin

## Abstract

Hepatocellular carcinoma (HCC) is a lethal malignancy associated with drug resistance, resulting in a poor prognosis. High mobility group box 1 (HMGB1) is a chromatin-binding protein that regulates HCC progression. The overexpression of HMGB1 has been found to promote tumorigenesis and drug resistance. In this study, we aimed to investigate the role of HMGB1 expression in tumorigenesis and metastasis and its impact on sorafenib and oxaliplatin resistance. Tissue samples from patients with HCC (*n* = 48) were subjected to immunohistochemistry. The expression of HMGB1 was correlated with clinical pathology parameters. Moreover, the HCC cell line HuH-7 was used to study the regulatory effect of HMGB1 on cell proliferation, cell adhesion, migration, and invasion by using the siRNA (small interfering RNA) silencing method. Furthermore, drug challenges were performed to determine the effect of HMGB1 on the sensitivity to chemotherapeutic drugs (sorafenib and oxaliplatin). HMGB1 was significantly overexpressed in tumor tissues, highlighted by the expression increment in patients with M1 advanced metastasis tumors with immunoreactivity scores 2.61 and 6.50 for adjacent and tumor tissues, respectively (*p*-values = 0.0035). The involved mechanisms were then described through the suppression of HCC cell adhesion, migration, and invasion by HMGB1 silencing. Notably, the inhibition of HMGB1 expression promoted sorafenib/oxaliplatin sensitivity in the HCC cell line by increasing the cell toxicity by about 13–18%. Our study demonstrated that HMGB1 shows potential as a promising biomarker and a target for HCC treatment that is involved in tumorigenesis, metastasis, and chemo-drug resistance.

## 1. Introduction

HCC is one of the most fatal malignancies worldwide, particularly in Asia and Africa [1]. The two-year survival rate reportedly decreases by half in patients with HCC, especially older patients, despite the introduction of transarterial chemoembolization (TACE) therapy. Interestingly, advanced-stage HCC has been observed in patients who have undergone TACE and surgery [2]. Sorafenib (a multiple tyrosine kinase inhibitor) and oxaliplatin remain the first- and second-line drugs, respectively, in conventional toxicity agent therapy and chemotherapy. Owing to its anti-proliferative, anti-angiogenic, and anti-invasion properties, sorafenib has been exploited in TACE, resulting in improved progression-free survival compared to TACE alone [3,4,5]. However, only 30% of patients with HCC experience benefits, and sorafenib resistance has been confirmed within 6 months among patients with HCC [6]. Oxaliplatin resistance has also been reported as a consequence of combination chemotherapy [7]. Consequently, overcoming drug resistance is indispensable to promote cancer progression-free survival in patients with HCC.

HMGB1 is the nucleosome-interacting protein that facilitates DNA transcription. However, the hyperacetylation of lysine residues promotes its translocation into the cytosol. HMGB1 is a damage-associated molecular pattern released or secreted following cell death or injury, respectively [8]. HMGB1 can activate molecular mechanisms through specific cell membrane receptors, causing abundant nearby cell responses. Recently, HMGB1 overexpression was shown to promote HCC cell invasion and oxaliplatin resistance [9]. Extracellular HMGB1 was found to induce HCC progression and metastasis by upregulating toll-like receptor 4 (TLR4) [10]. Interestingly, extracellular HMGB1 has been implicated as a crucial protein to enhance HCC progression by activating the receptor for advanced glycation end product (RAGE)/miR-21/OCT4/CD44 cascades [11,12]. Despite considerable evidence confirming that HMGB1-stimulated HCC progression results in poor prognosis, the association between HMGB1 expression and metastatic staging, including drug resistance, remains limited.

In this study, we aimed to investigate the role of HMGB1 expression in tumorigenesis and metastasis and its impact on sorafenib and oxaliplatin resistance. Tissue samples from 48 patients with HCC were embedded in paraffin and stained with an HMGB1 antibody. HMGB1 was also knocked down in the HCC cell line HuH-7 and treated with sorafenib and oxaliplatin to examine drug sensitivity. Herein, we found that HMGB1 was overexpressed in tumor tissues when compared with that in nearby normal tissues. Interestingly, HMGB1 knockdown in the HCC cell line increased sensitivity to sorafenib and oxaliplatin when compared with that in controls. Based on our findings, HMGB1 may be involved in tumorigenesis, metastasis, and drug resistance. Thus, HMGB1 is a promising marker of HCC progression and a potential target for HCC treatment.

## 2. Results

### 2.1. Overexpression of HMGB1 in Tumor Tissues and Patients with Metastasis

To determine the clinical significance of HMGB1 in tumors, the expression level of HMGB1 was determined using immunohistochemistry. Forty-eight parafilm-embedded HCC sections were stained with an anti-HMGB1 antibody and counterstained with hematoxylin for nuclear staining. The burden on the tumor and adjacent tissue was identified, and the immune reactive score (IRS) was evaluated independently by two pathologists. Herein, the HMGB1 expression was significantly increased in the tumor area when compared with that in its adjacent counterpart (Figure 1A,B). Although clinicopathologic parameters, such as the tumor size, lymph node metastasis, tumor extension, and vascular invasion, were not significantly associated with HMGB1 expression, the expression of HMGB1 strongly and positively correlated with the metastatic status of patients with HCC. The M1 patients exhibited a significant increase in HMGB1 protein expression when compared with the M0 patients (Table 1 and Figure 1C). Thus, HMGB1 appears to be a crucial element of tumor mass and may be involved in tumorigenesis and metastasis.

### 2.2. HMGB1 Plays a Crucial Role in HCC Cell Proliferation and Metastasis

To demonstrate that HMGB1 contributes to HCC metastasis, we performed in vitro experiments using the HCC cell line HuH-7. Cells were transfected with siRNA to downregulate the expression of HMGB1, which was detected using immunoblotting and RT-qPCR. The results showed that HMGB1 was knocked down, as indicated by slight expression at the protein level and significantly suppressed expression at the mRNA level (Figure 2A,B). Interestingly, cells with siRNA-mediated downregulation exhibited significant suppression of HCC cell proliferation on day 3 when compared with silencer negative control siRNA (siControl) (Figure 2C).

Cancer cell metastasis involves cell adhesion, migration, and invasion. The HuH-7 activity was determined following the downregulation of HMGB1. We demonstrated that knockdown of HMGB1 significantly suppresses HCC adhesion, migration, and invasion through the extracellular matrix (Figure 3A–C). These results indicated that HMGB1 is an important regulator of HCC cell metastasis.

### 2.3. Inhibition of HMGB1 Increases Drug Sensitivity

HMGB1 confers drug resistance and results in unfavorable HCC treatment. Herein, we determined the effectiveness of sorafenib and oxaliplatin in HuH-7 cells with knocked-down HMGB1 expression. The HCC cells were treated with sorafenib and oxaliplatin for 24 and 48 h, and cell viability was measured and calculated as the half maximal inhibitory concentration (IC_50_) of each drug (Figure 4A). The cells were then rechallenged with these drugs, either in the control or HMGB1 knockdown groups, and cell viability was determined. At the same concentration of sorafenib (2.73 µM) and oxaliplatin (148.5 µM), HMGB1 expression significantly enhanced the cell toxicity of drugs when compared with the control group, except for the 48 h treatment with oxaliplatin.

## 3. Discussion

HMGB1 is reportedly involved with various tumor mechanisms, including tumor growth [11], metastasis [13], angiogenesis [14], and chemoresistance [15]. Especially in HCC, HMGB1 has been found to be associated with chronic liver injury and hepatocarcinogenesis [16]. In the current study, we emphasized the importance of HMGB1, which was markedly overexpressed in tumor tissues when compared with that in normal hepatocytes (Figure 1). Our findings are consistent with those of a previous report showing that HMGB1 is highly expressed in HCC cell lines [17] and tissues [18]. HMGB1 participates in hepatocarcinogenesis via various mechanisms; for example, the interaction of HMGB1 and its specific receptor, RAGE, promotes the tumorigenesis and proliferation of HCC [19,20]. Moreover, the binding of HMGB1 to TLR-9 receptor induce tumor growth under hypoxic conditions [21], which was emphasized by our demonstration that the downregulation of HMGB1 reduced HCC cell growth. In addition, our findings demonstrated that HMGB1 overexpression was associated with distant metastasis (Figure 1C and Table 1). Upon analyzing HMGB1 localization, we found that HMGB1 was strongly expressed in the enlarged tumor group (T4) and in tumors with perforation in the visceral peritoneum group (Appendix A). The subcellular localization of HMGB1 was analyzed but did not differ significantly between tumor vs. normal tissues or distinct metastasis status. Reportedly, HMGB1 is more highly expressed in the nucleus and cytoplasm of colorectal cancer than in normal colorectal tissue, but there was no significant difference in the distinct metastasis analysis [22].

Subsequently, the effect of HMGB1 on tumor progression was confirmed by an in vitro study, which revealed that HMGB1 knockdown could suppress the proliferation, adhesion, migration, and invasion abilities of HuH-7 cells. The role of HMGB1 in tumor metastasis has been demonstrated in several cancer types, including lung [13], colon [22], and liver [18] cancers. The stimulation of the TLR4 and RAGE signaling pathways by HMGB1 may promote cancer invasion and metastasis via caspase-1 activation and a cascade of inflammatory responses [17]. The interaction of extracellular HMGB1 on TLR4 was shown to facilitate the stimulation of adaptor proteins, including MyD88, TIRAP, TRIF, and TRAM, subsequently activating the nuclear factor kappa-light-chain-enhancer of activated B cells (NF-κB). In addition, RAGE activation could promote essential growth signaling pathways, including PI3K and ERK, which directly promote cancer cell proliferation and invasion. Moreover, RAGE signaling was found to be associated with NF-κB and MAPK, which enhance tumor progression by inducing the production of chemokines and cytokines [23]. The inhibition of p38/JNK-HMGB1 and HMGB1/NF-κB/STAT3 signaling attenuated the invasion and metastasis abilities of HCC cells [24,25]. Therefore, our study emphasizes the effectiveness of HMGB1 as a prognostic marker in patients with HCC. In particular, we present immunohistochemistry results, which can be conveniently combined with conventional clinical pathology. Moreover, extracellular HMGB1 circulates in the bloodstream and other body fluids, such as joint synovial fluid, urine, bronchoalveolar lavage fluid, and cerebrospinal fluid. Elevated serum HMGB1 levels were found to correlate with tumor progression and metastasis in non-small cell lung and gastric cancers [26,27]. The validation of the clinical diagnosis is urgently needed for the early detection of HCC or for monitoring metastasis without tumor dissection.

Drug resistance presents a considerable challenge in the treatment of HCC. In 2007, the U.S. Food and Drug Administration (USFDA) approved sorafenib as a therapeutic agent to treat HCC, which was found to improve the median overall survival and disease control rates compared with the placebo group [28]. Sorafenib functions as a multitarget tyrosine kinase inhibitor that inhibits angiogenesis and proliferation. However, only ~30% of patients experience benefits following treatment with sorafenib and frequently develop drug resistance within six months [29]. HMGB1 promotes HCC resistance to sorafenib by stimulating the downstream MAPK cascade through Erk1/2 phosphorylation, activating autophagy, and eventually inhibiting cancer cell apoptosis [30]. Hence, in this study, we demonstrated that the suppression of HMGB1 significantly sensitizes HCC cells to sorafenib treatment (Figure 4). In addition, we determined the effects of the second-line chemotherapeutic drug oxaliplatin, frequently used to enhance the efficacy of sorafenib. The inhibition of HMGB1 expression increased the cytotoxicity of oxaliplatin, even after a short treatment time (Figure 4). Targeting HMGB1 by siRNA transfection reduces the expression of endogenous HMGB1, which plays a crucial role in autophagy-associated drug resistance by binding to beclin 1 and eventually stimulating autophagosome formation [30]. Hence, the specific molecular mechanism of HMGB1 knockdown may involve the modulation of cell death by attenuating cell autophagy and increasing cell apoptosis. Accordingly, our results emphasize the potential of HMGB1 as a target for drug sensitization. Several inhibitors of HMGB1 have been identified, including ethyl pyruvate, which inhibits HMGB1 expression and cytoplasmic localization via the inhibition of HMGB1 acetylation, and glycyrrhizin, which directly binds to the Box A and B structure and prevents interaction with its receptors. In this study, we demonstrated the role of HMGB1 in HCC metastasis and drug sensitivity using siHMGB1 knockdown. For long-term treatment, HMGB1 inhibitors should be considered to control their modes of action and side effects. Although several studies have shown that ethyl pyruvate can induce tumor suppression, evidence supporting its effect on chemotherapy drug sensitivity is lacking [31,32,33]. Therefore, further investigations exploring the combination of ethyl pyruvate with sorafenib and oxaliplatin should be undertaken for potential clinical use.

Considering the limitations of this study, a larger number of tissue samples should be considered to generate further robust evidence. Moreover, long-term observations will determine the survival fraction of low/high HMGB1 expression, which could reasonably describe the role of HMGB1 as a prognostic biomarker. In addition, the underlying mechanistic relationships between HMGB1, cancer metastasis, and drug resistance warrant further investigation, either through molecular signaling or concept confirmation in animal models.

## 4. Materials and Methods

### 4.1. Cell Culture

The HCC cell line Huh-7 was kindly gifted by Dr. Krai Daowtak, Naresuan University, Phitsanulok, Thailand. Cells were maintained in Dulbecco’s Modified Eagle Medium (Gibco, Thermo Fisher Scientific, New York, NY, USA) containing 10% fetal bovine serum (FBS), 1% antibiotic–antimycotic (10,000 U/mL of penicillin, 10,000 µg/mL of streptomycin, and 25 µg/mL of Gibco Amphotericin B) (Gibco, Thermo Fisher Scientific). The cells were cultured at 37 °C under 5% CO_2_ and 95% ambient air.

### 4.2. Patients and Tissue Samples

Tissue samples were collected from 48 patients with HCC. All participants underwent initial surgical intervention at the Affiliated Naresuan University Hospital, Tha-pho, Phitsanulok, Thailand, from 2017 to 2023. The study was approved by the Human Research Ethics Committee of Naresuan University, according to the Declaration of Helsinki, Belmont report, CIOMS guidelines, and ICH-GCP (IRB No. P1-0006/2567). Patient medical records were used to assess the clinicopathological characteristics of all patients. The attributes and distribution of the HCC cohort are shown in Table 1. The study methods followed the guidelines and regulations established by the affiliated Naresuan University Hospital.

### 4.3. Immunohistochemistry Staining

Paraffin-embedded tissues were stained in a 2-step plus Poly-HRP Anti-Rabbit IgG Detection System (ElabScience^®^, Houston, TX, USA). Briefly, the paraffin sections were dewaxed and rehydrated. The slides were incubated with 3% hydrogen peroxide (H₂O₂) for 10 min, followed by washing with phosphate-buffered saline (PBS) for 2 min; this was repeated three times. Normal Goat Blocking Buffer was added, and cells were then incubated at 37 °C for 30 min. Anti-rabbit HMGB1 (ElabScience) was added and incubated for 2 h, followed by washing with PBS for 2 min, three times. Polyperoxidase-anti-Rabbit IgG was applied and incubated at room temperature or 37 °C for 20 min, followed by washing with PBS. The 3,3′-diaminobenzidine (DAB) Working Solution was added to the sections, which were then washed with DI water to terminate the chromogenic reaction before counterstaining with hematoxylin.

The tissue sections were independently assessed and scored by two pathologists. Staining intensity was rated on a four-point scale: 0 (negative), 1 (weak), 2 (medium), or 3 (strong). The percentage of positive cells was evaluated using a four-point scale: 0 (1–25%), 1 (26–50%), 2 (51–75%), or 3 (76–100%). To determine the IRS, the staining intensity score was multiplied by the percentage of positive cells.

### 4.4. siRNA Transfection

To inhibit the expression of HMGB1 in Huh-7 cells, the cells were transfected with siRNA. Huh-7 cells were seeded into 96-well plates at a density of 5 × 10^3^ cells/well, 24-well plates at 3 × 10^4^ cells/well, or 6-well plates at 4 × 10^5^ cells/well, and incubated overnight until the cells reached 70–90% confluency. HMGB1 expression was knocked down using Silencer^®^ Pre-designed siRNA (siHMGB1) (Thermo Fisher Scientific, Life Technologies Corporation, Carlsbad, CA, USA) compared with siControl as the negative control (Thermo Fisher Scientific) using Lipofectamine™ 3000 Reagent (Invitrogen, Thermo Fisher Scientific, New York, NY, USA) according to the manufacturer’s instructions. Briefly, preparation of siRNA master mix by adding of siRNA to P3000™ reagent and cell culture medium to yield final concentration of siHMGB1 at 0.2, 1 and 5 µg in 96/24 and 6-well cell culture plate, respectively. The master mix was then mixed with Lipofectamine™ 3000 and incubated before subsequently transfection to Huh-7 cells for 6 h. The transfection efficacy was confirmed by determining the expression of HMGB1 at 24 h, using Western blotting and reverse transcription-quantitative PCR (RT-qPCR).

### 4.5. Cell Adhesion Assay

Next, we determined the adhesion ability of Huh-7 cells after knocking down HMGB1 expression compared with the negative control group. The wells of 96-well plates were coated with Corning^®^ Matrigel^®^ Basement Membrane Matrix (Corning, Life Sciences, Waltham, MA, USA) and incubated overnight. The cells were seeded at density 8 × 10^3^ cells/well into the wells coated with Matrigel, incubated at 37 °C under 5% CO_2_ and 95% air for 30 min, and then washed with PBS three times to remove non-adhering cells. Adherent cells were stained with 0.5% Crystal Violet in 20% methanol for 30 min. The stained cells were eluted with 70% methanol, and the absorbance was measured at 570 nm using a microplate spectrophotometer. The absorbance values were used to calculate the percentage of cell adhesion using the following formula:$$\%\;\mathrm{Cell}\;\mathrm{Adhesion}\;=\frac{O.D.\;570\;\mathrm{of}\;\mathrm{Knockdown}\;\mathrm{HMGB}1\;\mathrm{group}}{O.D.\;570\;\mathrm{of}\;\mathrm{Negative}\;\mathrm{Control}\;\mathrm{group}}\times 100$$

### 4.6. Cell Scratch Assay

A cell scratch assay was performed to assess the migratory ability of Huh-7 cells. The cells were seeded into a 6-well cell culture plate at a density of 4 × 10^5^ cells/well and incubated overnight at 37 °C under 5% CO_2_ and 95% air until 90–100% confluency was achieved. After knocking down the expression of HMGB1, cells were scratched at the middle of the well using a 200 µL micropipette tip, followed by incubation at 37 °C under 5% CO_2_ and 95% air. The migratory ability of cells was visualized and captured at 0, 24, and 48 h using Celloger^®^ Mini Plus (Gangnam, Seoul, Republic of Korea).

### 4.7. Cell Invasion Assay

The cell invasion assay was performed in a trans-well cell culture chamber. The upper chamber inserts in the 24-well plates were coated with Corning Matrigel Basement Membrane Matrix (Corning) and incubated overnight at 37 °C under 5% CO_2_ and 95% air. Subsequently, the cells were seeded into the upper chamber of trans-well inserts at density 5 × 10^4^ cells/well, and the complete DMEM medium was added into the lower chamber and incubated for a further 24 h at 37 °C under 5% CO_2_. The cell culture medium was then removed, and the upper chamber was gently cleaned using a cotton swab. The invading cells were fixed with 4% formaldehyde for 10 min and stained with 0.5% Crystal Violet in 20% methanol for 15 min. The remaining stain was removed using tap water, and the sample was allowed to dry. The invading cells were visualized and counted using an inverted microscope. The % invasion was calculated using the following formula:$$\%\;\mathrm{Cell}\;\mathrm{Invasion}=\left(\frac{\mathrm{Number}\;\mathrm{of}\;\mathrm{cells}\;\mathrm{in}\;\mathrm{Knockdown}\;\mathrm{HMGB}1\;\mathrm{group}}{\mathrm{Number}\;\mathrm{of}\;\mathrm{cells}\;\mathrm{in}\;\mathrm{Control}\;\mathrm{group}}\right)\times 100\;$$

### 4.8. Drug Challenge Assay

The effect of HMGB1 on drug resistance was examined in HuH-7 cells. Sorafenib and oxaliplatin were used in all the experiments. Huh-7 cells were seeded into 96-well plates at density 5 × 10^3^ cells/well, followed by incubation overnight at 37 °C under 5% CO_2_ and 95% air. The cells were treated with different concentrations of chemotherapeutic drugs and incubated for 24 and 48 h. The IC_50_ values of sorafenib (2.73 µM) and oxaliplatin (148.5 µM) were determined using the MTT assay and were used for further treatment of HMGB1 knocked-down Huh-7 cells compared with the negative control group to determine the drug sensitivity of the cells.

### 4.9. Statistical Analysis

Data analyses were performed using GraphPad Prism version 10.4.0 (GraphPad Software, Inc., La Jolla, CA, USA). The results are expressed as mean ± standard deviation. Group differences were evaluated using unpaired *t*-tests and analysis of variance (ANOVA), with significance set at *p* < 0.01.

## 5. Conclusions

We demonstrated that HMGB1 is a crucial element in tumorigenesis and tumor metastasis, both in vitro and in clinical settings. Thus, HMGB1 could be a potential prognostic biomarker for distant tumor metastasis. In the context of cancer metastasis and drug resistance, HMGB1 should be considered a target therapy for HCC treatment, as evidenced in the current study and previous reports. Therefore, HMGB1 is a promising target for HCC treatment and diagnosis.

## Figures and Tables

**Figure 1 ijms-26-03491-f001:**
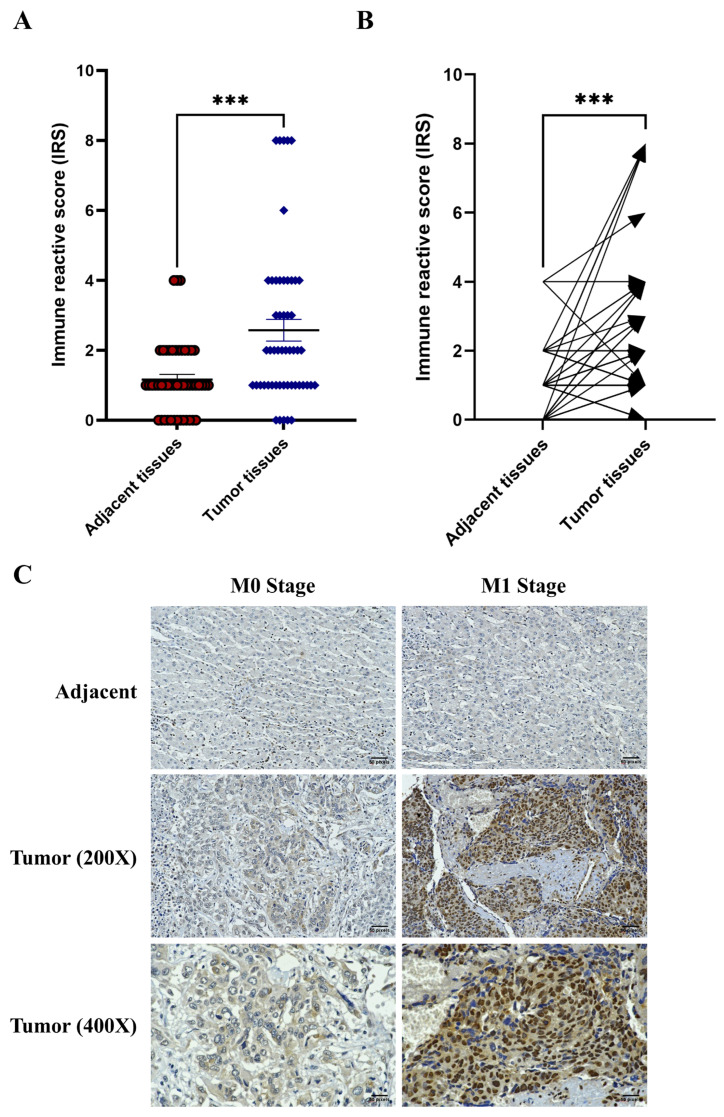
HMGB1 is highly expressed in tumor tissues and patients with metastasis. Paraffin-embedded tumor biopsies were evaluated for HMGB1 expression by performing immunohistochemistry, counterstained with hematoxylin and photographed (200–400×) (**C**). IRS was evaluated and compared between tumor and adjacent tissues represented by either overall grouping (**A**) or individual patients (**B**). The data are presented as the means ± SEM of forty-eight tumor IRSs. Significant differences are indicated by *** *p* < 0.001.

**Figure 2 ijms-26-03491-f002:**
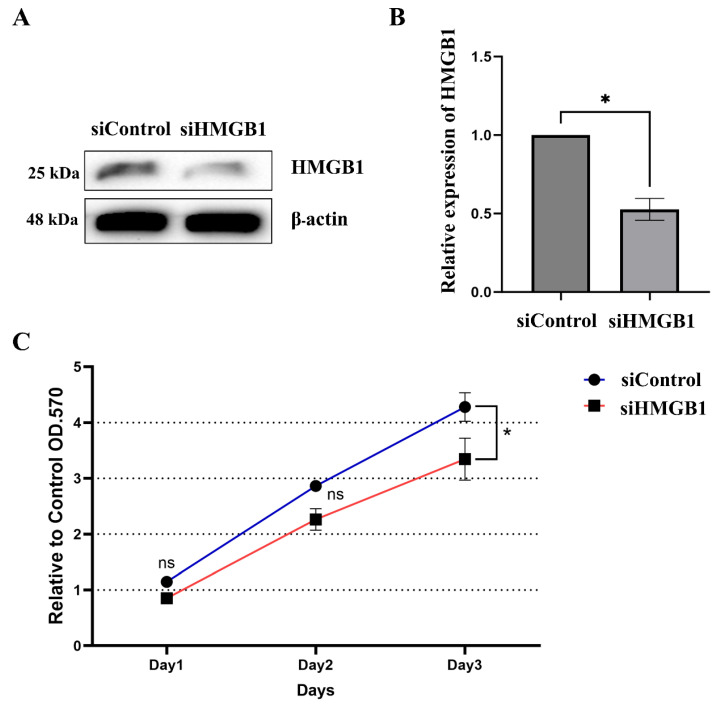
HBMG1 knockdown weakly impacts cancer cell proliferation. siRNA transfection was performed to downregulate HMGB1 expression in HuH-7 cells. The expression level of HMGB1 was confirmed by performing immunoblotting (**A**) and RT-qPCR (**B**). Growth curve analysis of either control or knockdown HMGB1 groups was measured using the MTT assay (**C**). All quantitative data are shown as the mean ± SEM (*n* = 6). Significant differences are indicated by * *p* < 0.05 compared with the control group. ns, not significant; siControl, silencer negative control siRNA; siHMGB1, Silencer^®^ pre-designed siRNA to HMGB1.

**Figure 3 ijms-26-03491-f003:**
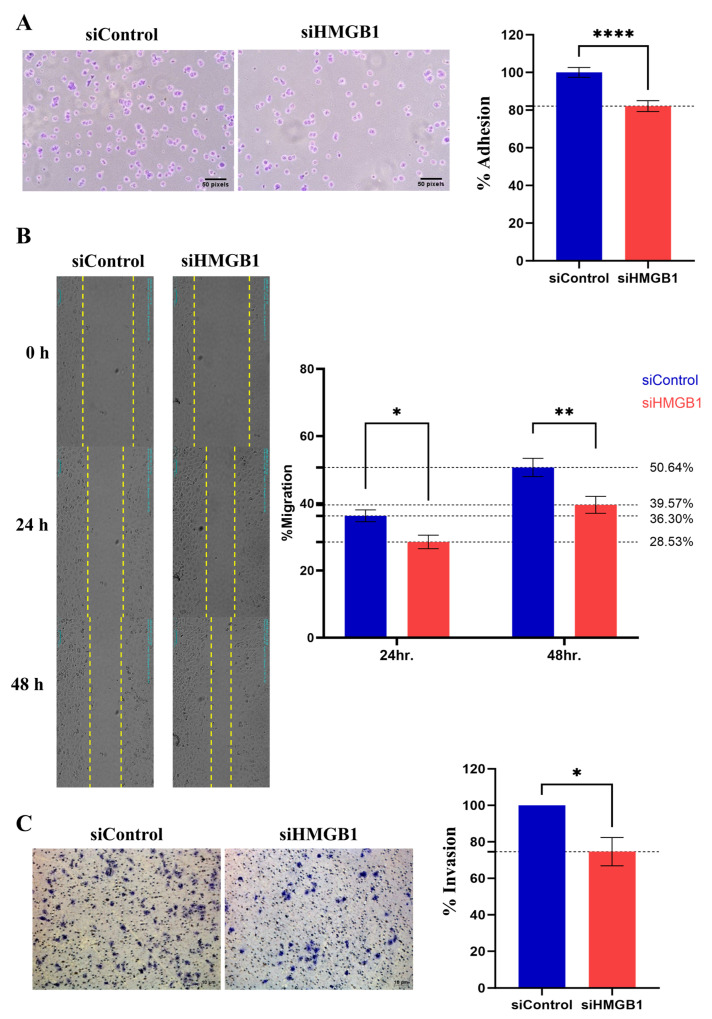
Knockdown of HMGB1 expression attenuates cancer cell metastasis. HuH-7 cells and siHMGB1 treatment were evaluated to determine their metastasis ability, including cell adhesion (**A**), cell migration using the cell scratch assay, the yellow line indicated for wound margin (**B**), and cell invasion using the trans-well assay (**C**). All quantitative data are shown as the mean ± SEM (*n* = 6). Significant differences are indicated by * *p* < 0.05, ** *p* < 0.01 and **** *p* < 0.0001, compared with the control group. The results were photographed under an inverted microscope at 100× magnification. siControl, silencer negative control siRNA; siHMGB1, Silencer^®^ pre-designed siRNA to HMGB1.

**Figure 4 ijms-26-03491-f004:**
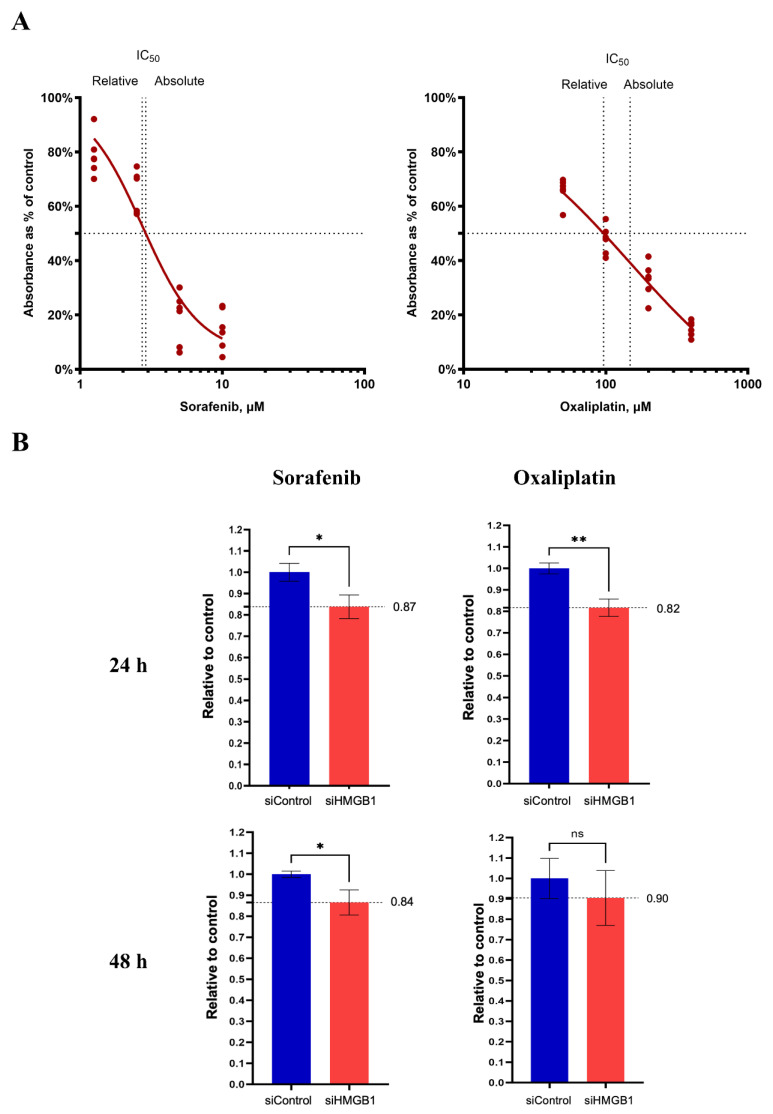
Downregulation of HMGB1 expression enhances the sensitivity of HCC treatment drugs. After treatment with sorafenib and oxaliplatin, the viability of HuH-7 cells was measured using the MTT assay, and the IC_50_ curve with drug concentration was determined (**A**). HCC cells were then challenged with drugs at the IC50 concentrations; the inhibitory ability was shown as relative to the transfection control group (**B**). All quantitative data are shown as the mean ± SEM (*n* = 6, 2 times independently). HCC, hepatocellular carcinoma. Significant differences are indicated by * *p* < 0.05, ** *p* < 0.01, compared with the control group. IC_50_, half maximal inhibitory concentration; ns, not significant.

**Table 1 ijms-26-03491-t001:** Association of HMGB1 expression with clinicopathologic parameters in patients with hepatocellular carcinoma.

Variables	*n* (%)	Mean IRS	*p*-Value
Sex			
Male	35 (73%)	3.029	0.6982
Female	13 (27%)	2.692	
Age			
<60	13 (27%)	3.154	0.7323
≥60	35 (73%)	2.857	
T staging			
T1	26 (55%)	3.269	0.3416
T2	16 (33%)	2.000	
T3	1 (2%)	3.000	
T4	5 (10%)	4.200	
M Staging			
M0	44 (92%)	2.614	*0.0035* *
M1	4 (8%)	6.500	
Tumor extension			
Confined to liver	43 (90%)	2.791	0.2610
Involves visceral peritoneum	5 (10%)	4.200	
Vascular invasion			
Not identified	30 (62%)	3.400	0.1166
Present	18 (38%)	2.167	

IRS, immunoreactivity score. Significant differences are indicated by * *p* < 0.05.

## Data Availability

The original contributions presented in this study are included in the article/Appendix A. Further inquiries can be directed to the corresponding author.

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
