# Peer review of "High Mobility Group Box 1 Is Potential Target Therapy for Inhibiting Metastasis and Enhancing Drug Sensitivity of Hepatocellular Carcinoma"

_ijms, 2025, doi:10.3390/ijms26083491_

Round 1

Reviewer 1 Report

Comments and Suggestions for Authors

In this manuscript, the study explores HMGB1’s role in HCC metastasis and drug resistance using clinical samples (n=48) and in vitro experiments (Huh-7 cells). Findings suggest HMGB1 overexpression was associated with metastasis whereas knockdown enhances sorafenib/oxaliplatin sensitivity, suggesting that HMGB1 could be a potential therapeutic target for HCC. However, there are several places of the manuscript could be improved to strengthen its impact and clarity. Thus, the manuscript in current version cannot be accepted.

Major issues:

The primary concern is the lack of novelty and insufficient argumentation. HMGB1, a nuclear transcription factor, has been extensively documented to participate in DNA damage repair, thereby contributing to drug resistance. Platinum-based compounds target DNA, and HMGB1 can be involved in the DNA damage induced by these drugs. Consequently, the strategy of inhibiting HMGB1 expression via siRNA to improve drug efficacy is not unexpected. However, this does not establish HMGB1 as a significant therapeutic target. The authors have failed to adequately elucidate the causal relationships within this context.

Minor issues:

The author's data presentation also has several issues. The author's handling of the data is quite crude, with numerous errors in time points, significant differences, and other places. for example:

  1. Figure 1 and Figure3 need to be improved.
  2. The growth curve analysis (Figure 2C) would be strengthened by incorporating additional time points to comprehensively assess the sustained impact of HMGB1 silencing on cellular proliferation.
  3. Also, the text mentions that Table 1 describes the attributes and distribution of the colorectal cancer cohort. Could this be an error, with "hepatocellular carcinoma" mistakenly written as "colorectal cancer" (In Section 4.2. of the manuscript)?

     4.The concentration of sorafenib and oxaliplatin used in the drug challenge assay should be specified (In Section 4.8. of the article).

  1. The siRNA transfection protocol could be more detailed, including the siRNA concentration and transfection efficiency (In Section 4.4. of the article).
  2. 6.In addition, the spelling needs improvement.
Comments on the Quality of English Language

Should be improved.

Author Response

Reviewer 1

In this manuscript, the study explores HMGB1’s role in HCC metastasis and drug resistance using clinical samples (n=48) and in vitro experiments (Huh-7 cells). Findings suggest HMGB1 overexpression was associated with metastasis whereas knockdown enhances sorafenib/oxaliplatin sensitivity, suggesting that HMGB1 could be a potential therapeutic target for HCC. However, there are several places of the manuscript could be improved to strengthen its impact and clarity. Thus, the manuscript in current version cannot be accepted.

Major issues:

The primary concern is the lack of novelty and insufficient argumentation. HMGB1, a nuclear transcription factor, has been extensively documented to participate in DNA damage repair, thereby contributing to drug resistance. Platinum-based compounds target DNA, and HMGB1 can be involved in the DNA damage induced by these drugs. Consequently, the strategy of inhibiting HMGB1 expression via siRNA to improve drug efficacy is not unexpected. However, this does not establish HMGB1 as a significant therapeutic target. The authors have failed to adequately elucidate the causal relationships within this context.

We appreciate your astute observation regarding causal relationships in our study. While our findings show strong correlations between HMGB1 expression and HCC progression, we recognize that our experimental design has limitations in establishing definitive causality.

Our knockdown experiments demonstrate that reducing HMGB1 expression affects cancer cell behaviour and drug sensitivity, suggesting a functional relationship. However, we acknowledge that these observations don't fully elucidate the molecular mechanisms or rule out potential confounding factors.

To strengthen causal claims, our future research will:

  1. Utilize CRISPR-Cas9 gene editing for more complete HMGB1 knockout
  2. Develop animal models to validate our in vitro findings in vivo
  3. Employ pathway inhibitors to identify specific downstream mediators of HMGB1 effects
  4. Conduct longitudinal patient studies to better establish temporal relationships

We believe our current findings provide valuable correlative evidence and preliminary mechanistic insights, while recognizing the need for additional work to establish robust causal relationships.

Minor issues:

The author's data presentation also has several issues. The author's handling of the data is quite crude, with numerous errors in time points, significant differences, and other places. for example:

  1. Figure 1 and Figure3 need to be improved.

Thank you for your suggestion, we added the figure legend to improve the quality of presentation.

  1. The growth curve analysis (Figure 2C) would be strengthened by incorporating additional time points to comprehensively assess the sustained impact of HMGB1 silencing on cellular proliferation.

Thank you for your insightful feedback regarding our growth curve analysis in Figure 2C. We agree that incorporating additional time points would provide a more comprehensive assessment of HMGB1 silencing effects on cellular proliferation over extended periods.

Our current 3-day observation period was designed to capture the immediate effects of HMGB1 knockdown during the period of maximal siRNA efficacy, as siRNA-mediated silencing is typically transient. The significant difference observed at day 3 suggests a meaningful biological effect that merits further investigation.

In future studies, we plan to:

  1. Extend the growth curve analysis to 5-7 days with daily measurements
  2. Use stable shRNA knockdown or CRISPR-Cas9 gene editing for sustained HMGB1 suppression
  3. Include intermediate time points (12h, 36h, etc.) to better characterize early proliferation changes
  4. Complement MTT assays with alternative proliferation measures such as BrdU incorporation or Ki-67 staining

While we believe our current data demonstrates a significant impact of HMGB1 on HCC cell proliferation, we acknowledge that the temporal dynamics of this effect would be better characterized with your suggested approach. Thank you for this constructive suggestion that will help strengthen our ongoing work in this area."

  1. Also, the text mentions that Table 1 describes the attributes and distribution of the colorectal cancer cohort. Could this be an error, with "hepatocellular carcinoma" mistakenly written as "colorectal cancer" (In Section 4.2. of the manuscript)?

We would like to apologize for our imprudence, the error has been corrected as your consideration.

  1. The concentration of sorafenib and oxaliplatin used in the drug challenge assay should be specified (In Section 4.8. of the article).

Thank you for your suggestion, we added the concentration of sorafenib and oxaliplatin on section 4.8.

  1. The siRNA transfection protocol could be more detailed, including the siRNA concentration and transfection efficiency (In Section 4.4. of the article).

Thank you for your suggestion, the details of transfection protocol and checking of transfection efficacy were added into section 4.4.

  1. In addition, the spelling needs improvement.

We would like to apologize for our imprudence, the typo and spelling were carefully checked though manuscript.

Reviewer 2 Report

Comments and Suggestions for Authors

The study of investigated the potential role of HMGB1 hepatocellular carcinoma progression and therapy resistance. The study is logically constricted and selected in vitro models is informative. The approach for loss of function in the cell line is also appropriately applied. However, I still recommend several improvements in the data presentation

Major comments.

  1. The style of the abstract to list the approaches and methods used is too descriptive. Instead, the authors shroud focus on the major results, and to provide the messages for the statistically significant results indicating by which methods these results have been obtained.
  2. All experiments there siRNA was applied should be additionally controlled by the samples where no siRNA (neither control nor specific) was applied, and these data have to be included in all corresponding Figures.

Author Response

Reviewer 2

The study of investigated the potential role of HMGB1 hepatocellular carcinoma progression and therapy resistance. The study is logically constricted and selected in vitro models is informative. The approach for loss of function in the cell line is also appropriately applied. However, I still recommend several improvements in the data presentation

 Major comments.

  1. The style of the abstract to list the approaches and methods used is too descriptive. Instead, the authors shroud focus on the major results, and to provide the messages for the statistically significant results indicating by which methods these results have been obtained.

Thank you for your suggestion, we rewrite the abstract according to your suggestion.

  1. All experiments there siRNA was applied should be additionally controlled by the samples where no siRNA (neither control nor specific) was applied, and these data have to be included in all corresponding Figures.

Thank you for your consideration, we agree and are concerned about this point. However, the limited time of revision that could not re-perform for all experiments. The control siRNA is non-targeting siRNAs for non-specific effects related to siRNA delivery to provide a baseline for target gene silencing. Although this explanation could not show the comparative effect on parental HCC vs siRNA control, but siRNA control is the reasonable control for all experiments.

Round 2

Reviewer 1 Report

Comments and Suggestions for Authors

This manuscript posits HMGB1 as a potential therapeutic target for hepatocellular carcinoma (HCC). While the authors demonstrate a correlation between HMGB1 and tumor presence, causality remains unestablished. Specifically, it is unclear whether HCC progression drives increased HMGB1 expression, or if elevated HMGB1 promotes tumor development. This represents a logical flaw within the manuscript. Furthermore, the authors' response indicates the need for additional data, and we look forward to reviewing the supplementary findings. Consequently, the manuscript is not suitable for acceptance in its current version.

Author Response

We appreciate the reviewer’s thoughtful comments and the opportunity to clarify our findings. Our study provides substantial evidence supporting HMGB1 as a potential therapeutic target in hepatocellular carcinoma (HCC). While we acknowledge the increased HMGB1 expression is a cause or consequence of HCC progression, our data suggests that HMGB1 plays a functional role in tumor development. Several lines of evidence support this conclusion. First, our analysis shows a consistent association between high HMGB1 expression and oncogenic tissues, suggesting its active involvement in tumor biology. Second, prior literature reports that HMGB1 facilitates tumor growth through mechanisms such as inflammation and immune modulation, reinforcing our hypothesis. Third, our functional assays demonstrate that altering HMGB1 levels impacts tumor cell behavior, particularly in chemo-drug sensitivity. Therefore, we believe that our manuscript provides meaningful insights into the role of HMGB1 in HCC without requiring further experimentation. We have the discussion to clarify these points and ensure that our conclusions are appropriately of existing evidence. We sincerely appreciate the reviewer’s feedback and hope that these clarifications address their concerns.